# Cross-Validation of Predictive Equation for Cardiorespiratory Fitness by Modified Shuttle Walk Test in Adults with Schizophrenia: A Secondary Analysis of the CORTEX-SP Study

**DOI:** 10.3390/ijerph182111390

**Published:** 2021-10-29

**Authors:** Mikel Tous-Espelosin, Sonia Ruiz de Azua, Nagore Iriarte-Yoller, Pedro M. Sanchez, Edorta Elizagarate, Agurne Sampedro, Sara Maldonado-Martín

**Affiliations:** 1GIzartea, Kirola eta Ariketa Fisikoa Ikerkuntza Taldea (GIKAFIT), Society, Sports and Physical Exercise Research Group, Department of Physical Education and Sport, Faculty of Education and Sport-Physical Activity and Sport Sciences Section, University of the Basque Country (UPV/EHU), 01007 Vitoria-Gasteiz, Spain; mikel.tous@ehu.eus; 2Physical Activity, Exercise and Health Group, Bioaraba Health Research Institute, 01009 Vitoria-Gasteiz, Spain; 3Cibersam, Department of Neuroscience, University of the Basque Country (UPV/EHU), 48940 Leioa, Spain; sonia.ruizdeazua@ehu.eus; 4Refractory Psychosis Unit, Psychiatric Hospital of Alava, 01007 Vitoria-Gasteiz, Spain; nagore.iriarteyoller@osakidetza.eus (N.I.-Y.); pedromanuel.sanchezgomez@osakidetza.eus (P.M.S.); eduardojose.elizagaratezabala@osakidetza.eus (E.E.); 5Faculty of Medicine, University of Deusto, 48007 Bilbao, Spain; 6Department of Psychology, Faculty of Health Sciences, University of Deusto, 48007 Bilbao, Spain; a.sampedro@deusto.es

**Keywords:** assessment, equation for estimation, field test, peak oxygen uptake, validation

## Abstract

Cardiorespiratory fitness (CRF) can be direct or estimated from different field tests. The Modified Shuttle Walk Test (MSWT) is suitable for all levels of function, allowing a peak response to be elicited. Therefore, we aimed (1) to validate the equation presented in the original study by Singh et al. for evaluating the relationship between MSWT with peak oxygen uptake (VO_2peak_) in adults with schizophrenia (SZ), (2) to develop a new equation for the MSWT to predict VO_2peak_, and (3) to validate the new equation. Participants (N = 144, 41.3 ± 10.2 years old) with SZ performed a direct measurement of VO_2peak_ through a cardiopulmonary exercise test and the MSWT. A new equation incorporating resting heart rate, body mass index, and distance from MSWT (R^2^ = 0.617; adjusted R^2^ = 0.60; *p* < 0.001) performs better than the Singh et al. equation (R^2^ = 0.57; adjusted R^2^ = 0.57; *p* < 0.001) to estimate VO_2peak_ for the studied population. The *posteriori* cross-validation method confirmed the model’s stability (R^2^ = 0.617 vs. 0.626). The findings of the current study support the validity of the new regression equation incorporating resting heart rate, body mass index, and distance from MSWT to predict VO_2peak_ for assessment of CRF in people with SZ.

## 1. Introduction

Schizophrenia (SZ) is a chronic severe mental illness with an important bearing on the presence of cardiovascular risk (CVR) factors due to an unhealthy lifestyle, including lack of physical activity [1], smoking, substance abuse, and poor diet [2], along with adverse effects of medications [3] and social and economic factors [4].

Hence, according to different clinical guidelines, the control and assessment of CVR in SZ patients are recommended [5,6,7]. In this regard, a previous study has shown that people with SZ have a moderate CVR compared to a low CVR in healthy controls [8]. Therefore, any physical exercise program offered to SZ patients as a non-pharmacological co-adjuvant intervention should include a previous assessment. 

Besides the traditional parameters, cardiorespiratory fitness (CRF, i.e., the capacity of the cardiovascular and pulmonary systems to meet the oxygen demands of skeletal muscle during physical work) is considered a vital sign and has emerged as a modifiable risk factor to attenuate the risk of developing non-communicable diseases [9,10]. Thus, a poor CRF level has been associated with a higher increased mortality risk, independently of other clinical risk factors [9].

The measurement of CRF can be direct, expressed as maximum or peak oxygen consumption (VO_2max_ or VO_2peak_), or estimated from different ergometer or field tests [11]. The gold standard for directly measuring and assessing CRF is the cardiopulmonary exercise test (CPET), both in healthy and clinical populations [11]. However, even though in recent years the CPET has become a more feasible choice for CRF assessment, it is time-consuming, requires specialized and expensive laboratory facilities as well as expert personnel to supervise and is not widely available in many centers [12]. Therefore, the general population, and people with SZ in particular, rarely have the opportunity to perform this test, and as a result, CRF is the only major risk factor that is not regularly assessed in the clinical setting [13]. Thus, the recommendations suggest that when an exercise program is to be carried out, it should include the determination of CRF using at least predictive equations [11].

In this respect, when CPET is not feasible, CRF can be estimated using a variety of field tests by performing an exercise test with maximal effort to achieve high rates of perceived exertion and a percentage of an age-estimated maximum heart rate (HR) [14]. Accordingly, one of the most widely used field tests is the Modified Shuttle Walk Test (MSWT), which is considered suitable for all levels of function and allows a peak response to be elicited. Previous studies have assessed the association between the MSWT and VO_2peak_ in different populations (i.e., adolescents, sedentary, lung cancer, chronic obstructive pulmonary disease, obese women, pulmonary arterial hypertension, and primary hypertension), concluding that this field walk test is objective, safe, valid, effective, reliable, and highly predictive for the assessment of functional capacity in each of the populations examined [15,16,17,18,19,20,21,22,23,24]. Nevertheless, like the rest of the tests, it may not be valid in all populations, and therefore, an analysis of the properties of the test (i.e., validity, reliability, repeatability, and sensitivity) should be carried out [25]. Originally, Singh et al. [26] proposed an equation to assess functional capacity in patients with chronic obstructive pulmonary disease using a 12-level protocol Incremental Shuttle Walk Test, the previous version of MSWT [19] and, more recently Jurio-Iriarte et al. developed another one for people with primary hypertension and obesity using the MSWT [27]. However, to the best of our knowledge, no reports are available that have analysed the relationship between the MSWT and VO_2peak_ in a cohort of adults with SZ. Therefore, the aims of the present study were: (1) to validate the equation presented in the original study by Singh et al. for evaluating the relationship between MSWT with VO_2peak_ in adults with SZ, (2) to develop a new equation for the MSWT to predict VO_2peak_, and (3) to validate the new equation.

## 2. Materials and Methods

### 2.1. Study Participants

The CORTEX-SP study was conducted between May 2018 and June 2021 in Vitoria-Gasteiz (Basque Country, Spain). This is a secondary baseline analysis of the study comprising a total of 144 participants (CORTEX) aged between 18 and 65 years (41.3 ± 10.2 years), 118 men (81.9%), and 26 women (18.1%). All participants had a diagnosis of SZ according to DSM-5 F20.9 (*Diagnostic and Statistical Manual of Mental Disorders, Fifth Edition*). All the selection criteria (exclusion and inclusion criteria) and procedures for the CORTEX-SP study have previously been described in the primary analysis [8]. The study was approved by the Research Ethics Committee of the Basque Country (PI2017044), and written informed consent was obtained from all participants before any data collection.

### 2.2. Measurements

Body mass index (BMI) was calculated as the total body mass divided by height squared in meters (kg/m^2^). Waist and hip circumferences were taken, and the waist to hip ratio (WHR) was defined as waist circumference divided by hip circumference, both in centimeters. An ambulatory blood pressure (BP) monitoring recorder (ABPM) (6100 and 7100, Welch Allyn, New York, NY, USA) was used to measure BP for a whole day (24 h), through intervals of 30 min during the day and intervals of 60 min during the night. The variables considered from the ABPM measures were mean values of systolic BP, diastolic BP, and HR, all considered as resting values. 

The CRF was assessed through a CPET and MSWT [28] on separate days. The CPET was performed on an electronically braked Lode Excalibur Sport Cycle Ergometer (Groningen, the Netherlands) starting at 40 W with a gradual increment of 10 W each minute in ramp protocol. The expired gas was analyzed with a system (Ergo CardMedi-soft S.S, Belgium Ref. USM001 V1.0) that was calibrated before each test in order to determine VO_2peak_, which was defined as the highest oxygen uptake value attained toward the end of the test. Achievement of VO_2peak_ was assumed with the presence of two or more of the following criteria: (1) volitional fatigue (>18 on Borg scale), (2) peak respiratory exchange ratio (RER) ≥ 1.1, (3) achieving >85% of age-predicted maximum heart rate, and (4) failure of oxygen consumption (VO_2_) and/or HR to increase with further increases in work rate [29]. The MSWT consisted of walking/running up and down a 10 m corridor at an incremental speed, as previously described by Bradley et al. [28]. The test was finished when the participant (1) reached the end of level 15, (2) was too breathless to maintain the required speed, (3) was more than 0.5 m away from the cone when the beep sounded, (4) achieving >85% of age-predicted maximum HR, or (5) if the patient experienced chest pain or angina, dizziness, mental confusion, or extreme muscle fatigue [28,30].

### 2.3. Statistical Analysis

Statistical analyses were performed using the R software package. Descriptive statistics were performed on the baseline participants’ characteristics. The Ordinary Least Squares method was used to estimate the β parameters of the equation from the sample data available. Subsequently, multiple linear regression was used to generalize a model, and we determined which variables were the strongest predictors of VO_2peak_ through a variable selection algorithm. Forward stepwise linear regression was performed to test the effects of sex, age, body mass, BMI, WHR, systolic BP, diastolic BP, resting HR, peak HR at MSWT, and distance performed in the CPET, and to determine which variables are the strongest predictors of VO_2peak_. In the residual analysis of the regression model, the type of method used for the assessment of outliers was Bonferroni test, for autocorrelation was Durbin—Watson test, and for homoscedasticity was Non-constant Variance Score test. Finally, for the validation of the model, we decided to use the k-fold cross-validation method (k = 5). Statistical significance was set at *p* < 0.05.

## 3. Results

Descriptive data from the sample are presented in Table 1, and descriptive results have already been presented [8].

The formula of Singh et al. [26] in the present cohort was calculated as:VO_2peak_ = [6.271 + (0.021 − MSWT distance in meters)]

The residuals showed poorly centered values by analyzing the median (−0.811) and the minimum (−8.476) and maximum (15.071) ranges. Likewise, the predicted VO_2peak_ was significant yet of moderate strength (R^2^ = 0.57), explaining 57% of the variance (adjusted R^2^ = 0.57; *p* < 0.001) and indicating a standard error of the estimate (SEE) of 4.75 mL·kg^−1^·min^−1^. In the residual analysis of the regression model, the tests applied for the assessment of outliers (*p* = 0.122) and autocorrelation (*p* = 0.712) returned insignificant *p*-values. However, the *p*-value obtained by the homoscedasticity values (*p* = 0.012) (i.e., constant variance) was significant. 

To potentially improve the prediction of VO_2peak_, the present investigation used forward stepwise regression to identify other variables that may refine the prediction of VO_2peak_. In such a procedure, variables were added sequentially to the regression model as long as they significantly improved the predictive power of the model. Only the variables that contribute to the estimates using backward stepwise approach were used in the model. In the residual analysis of the regression model, the tests applied for the assessment of outliers (*p* = 0.248) and autocorrelation (*p* = 0.868) returned insignificant *p*-values. However, the *p*-value obtained by the homoscedasticity values (*p* = 0.001), (i.e., constant variance) was significant. A logarithmic transformation was performed to try to correct the heteroscedasticity. After the logarithmic transformation, homoscedasticity hypothesis was finally satisfied (*p* = 0.828). In this case, the VO_2peak_ was also significant, but still of moderate strength (R^2^ = 0.617), explaining, therefore, 60% of the variance (adjusted R^2^ = 0.60; *p* < 0.001) and indicating a SEE of 4.55 mL·kg^−1^·min^−1^ (Figure 1). 

The equation generated in this study to calculate VO_2peak_ (mL·kg^−1^·min^−1^) from the MSWT can be formulated as follows:VO_2peak_ = 20.168 − [0.226·BMI] − [0.064·Resting HR] + [0.019·Distance_MSWT_],
where BMI is expressed as kg/m^2^, resting HR (beats per minute) is measured before starting the MSWT with at least 5 minutes of resting time, and distance is measured in meters travelled in the MSWT.

Figure 2 illustrates the proportional bias (*p* < 0.001) with limits of agreement from −8.9 to 8.2 mL·kg^−1^·min^−1^. This indicates that using the new equation, the VO_2peak_ assessment of 95% of participants with SZ would range from 8.9 mL·kg^−1^·min^−1^ less to 8.2 mL·kg^−1^·min^−1^ more than their objective measure by CPET. Bland-Altman plot (Figure 2) shows that the biggest and smallest individual means of VO_2peak_ between the two tests correspond with the biggest limits of agreement on a proportional basis.

For the agreement and validity of the new equation, the summary of sample sizes created the five subsets (Table 2) and indicated the sample size of each of them (109, 109, 110, 108, 108). The R^2^ was 0.626 (Figure 1), compared with that of the original model, which was 0.617. Hence, it can be indicated that they are very similar and that the model is robust. Likewise, the average standard deviation of the subsets (SD = 0.092) was low, indicating a low variability between the models created. In summary, after the cross-validation method, it was concluded that the model was stable.

## 4. Discussion

In the present study, an analysis of the relationship between the MSWT and VO_2peak_ in a cohort of the SZ population was carried out. The main findings of this study were: (1) a new equation incorporating BMI, resting HR, and distance from MSWT performed better than the original Singh et al. equation [26] to estimate VO_2peak_ for the studied population, and (2) the *posteriori* cross-validation method confirmed the model’s stability. 

Previous studies have also presented correlations between measures of the shuttle walk test and CPET, explaining the 40.6% (healthy men) [16] and 57% (general surgical patients) [31] of the variance in VO_2peak_ and showing viability for the prediction of VO_2peak_. However, according to the present results, these equations, and even the equation of Singh et al. [26], may not be an appropriate method to estimate CRF through the MSWT for the assessment of functional capacity in people with SZ. Thus, the estimation equation by Singh et al. [26] showed that the residual values were poorly centered and only explained 57% of the variance (adjusted R^2^ = 0.57). The lack of precision could be because Singh et al. [26] generated the equation for patients with chronic airway obstruction, whose main limitation to exercise is dyspnea. In contrast to pulmonary patients, the performance of people with SZ is not usually limited by breathlessness but by exhaustion, low CRF, side effects relating to treatment (antipsychotic drugs), anhedonia, and lack of motivation [8,32]. Hence, among the variables obtained in the proposed models, BMI, and resting HR, together with the distance walked in the MSWT, proved to be the most relevant predictors. These variables explained the 60% of the variation in VO_2peak_ for the new equation, increasing the R^2^ and decreasing the SEE compared to Singh et al. equation (Figure 1). According to that, for the sample studied, the higher the BMI and resting HR, and the lower the distance walked at MSWT, the poorer the CRF. This can be explained by the inverse associations of high BMI, including an excess visceral adipose tissue (i.e., systemic inflammation) and elevated resting HR (i.e., increased sympathetic activity) with low CRF [33,34]. As an example of using the newly generated equation could be the following: if we provide two actual values observed for two participants with opposite CRF values included in this sample (Participant 1: measured VO_2peak_ = 40 mL·kg^−1^·min^−1^; BMI = 21.3 kg/m^2^; distance at MSWT = 1500 m; resting HR = 67 bpm. Participant 2: measured VO_2peak_ = 11 mL·kg^−1^·min^−1^; BMI = 46.6 kg/m^2^; distance at MSWT = 330 m; resting HR = 97 bpm), the estimated VO_2peak_ values would be 39.6 and 9.7 mL·kg^−1^·min^−1^, respectively, which corresponds with a very low error of estimate (0.4 and 1.3 mL·kg^−1^·min^−1^, respectively). 

One of the scientific goals of the present study was to predict an outcome (i.e., the VO_2peak_). Therefore, the metric by which to assess the quality of the prediction should be decided [35]. Previous studies have not validated the original formula [18,26] or there was a medium-validation result [27] without full support of the equation validity. In the current study, after using the cross-validation method, it was concluded that the model was stable with an R^2^ = 0.626 and had a high similarity with the newly generated equation (R^2^ = 0.617). Thus, the present results support the validity of the equation for routinely determining the CRF of SZ patients using the MSWT and the generated equation. However, we must be cautious with this statement since the validity of the MSWT-estimated VO_2peak_ is still moderate (60% of the variance). In this sense, in the clinical setting, when an accurate determination of CRF is critical, the CPET (with objective VO_2peak_ assessment) will remain the “gold standard”.

The current study has several strengths. Considering the difficulties involved in recruiting volunteers with a mental dysfunction, we could argue a relatively large sample (*n* = 144). Furthermore, the new equation could be a very useful and easy tool in the evaluation of this population, and the results obtained with this prediction are better compared to previous studies. There are, however, some limitations of this study that should be considered: (1) Symptoms affect each individual differently and have a direct bearing when assessing stress testing. (2) Cycle ergometer and MSWT tests are performed on different days and the motivation for exercise is highly variable from day-to-day in this population. (3) It has been observed that VO_2peak_ tends to be somewhat higher in those tests performed on a treadmill compared to those on a bike ergometer. Therefore, the SEE of the new equation (4.55 mL·kg^−1^·min^−1^) could be increased, since the direct measurement of VO_2peak_ has been done on the bike while the MSWT test is performed walking [36]. (4) The average age of the current cohort was 41.3 ± 10.2 years (range 18–65 years). This could limit the accuracy of the equation in younger or older individuals. (5) The ethnicity in the present study was predominantly white non-Hispanic (99%), suggesting that there may be potential ethnicity-related differences in the accuracy of the equation.

Finally, these results provide evidence for an easy, fast, and simple way to evaluate the CRF of people with SZ when objective measurements are not available. This would allow the necessary pre-design assessment of exercise in both clinical and non-clinical settings, and the promotion of exercise programs in this population.

## 5. Conclusions

In summary, the findings of this study support the validity of a new regression equation incorporating resting HR, BMI, and distance from MSWT to predict VO_2peak_ for assessment of CRF in people with SZ. However, when an accurate determination of functional capacity is required for diagnosis, clinical research, and exercise design, the direct measurement of VO_2peak_ will continue to be the “gold standard”.

## Figures and Tables

**Figure 1 ijerph-18-11390-f001:**
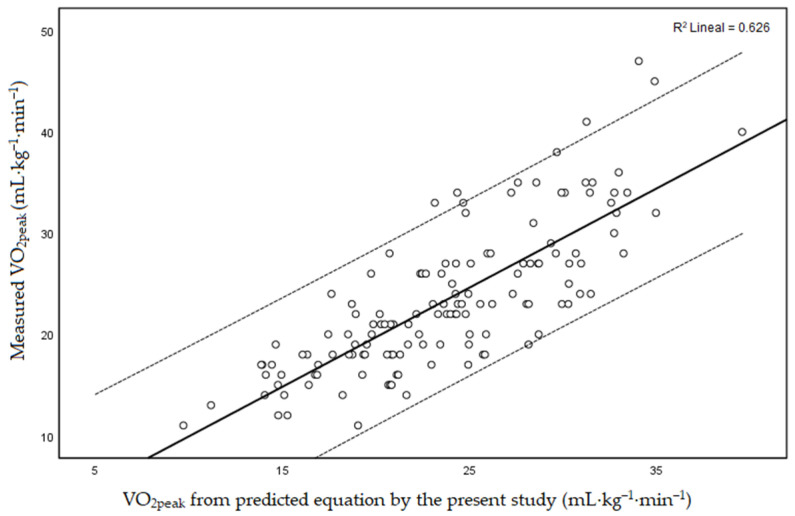
Relationship between measured VO_2peak_ and VO_2peak_ values from the predicted equation generated in the present study. The central line represents the linear regression line, and the flanking lines represent the 95% individual prediction intervals. CPET, cardiopulmonary exercise test; VO_2peak_, peak oxygen uptake.

**Figure 2 ijerph-18-11390-f002:**
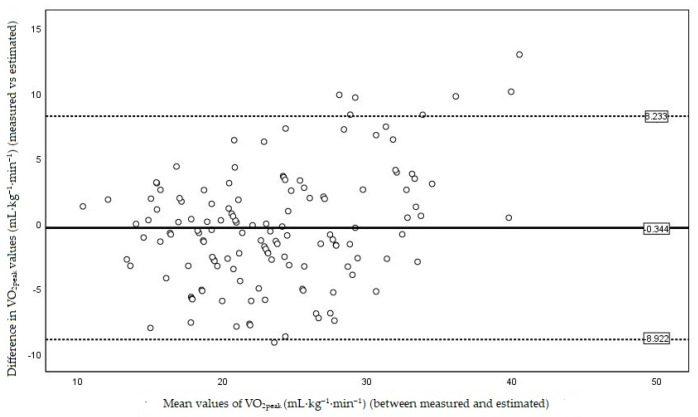
Bland and Altman plot. Intraindividual difference in VO_2peak_ (mL·kg^−1^·min^−1^) between two exercise tests (MSWT vs. CPET) plotted against intraindividual mean values of the exercise tests (MSWT and CPET). The central line represents the mean of the intraindividual differences, and the flanking lines represent the 95% limits of agreement.

**Table 1 ijerph-18-11390-t001:** Characteristics of the studied population. Values are means ± standard deviation or percentage (%).

Variables	*n* = 144
Age (yrs)	41.3 ± 10.2
Body mass (kg)	83.6 ± 16.4
BMI (kg/m^2^)	28.7 ± 7.4
Waist (cm)	96.6 ± 14.1
Hip (cm)	104.3 ± 9.5
WHR	0.93 ± 0.09
Resting systolic BP (mmHg)	116 ± 13
Resting diastolic BP (mmHg)	71 ± 8
Resting HR (bpm)	81 ± 11
Cigarette smoking (%)	66
Diabetes Mellitus (%)	7.6
Antipsychotic treatment (%) First-generation Second-generation Mixed	100 2.4 93.6 4.0
CPET variables	
Workload peak (W)	126 ± 37.7
Distance (km)	1.7 ± 0.9
HRpeak (bpm)	152.3 ± 19.7
VO_2peak_ (mL·kg^−1^·min^−1^)	23.3 ± 7.2
RER_peak_	1.2 ± 0.1
MSWT (m)	798.9 ± 265
HR_peak_ in MSWT (bpm)	153 ± 22.4

BMI, body mass index; WHR, waist to hip ratio; BP, blood pressure; HR, heart rate; CPET: cardiopulmonary exercise test; HR_peak_, peak heart rate; VO_2peak_, peak oxygen uptake; RER_peak_, peak respiratory exchange ratio; MSWT, modified shuttle walking test.

**Table 2 ijerph-18-11390-t002:** The evaluation indices for the 5-fold cross-validation.

Subsets	RMSE	R^2^	MAE
1	4.617	0.668	3.302
2	4.292	0.681	3.307
3	4.862	0.557	3.757
4	4.196	0.636	3.155
5	4.617	0.586	3.906

RMSE, Root mean square error; MAE, mean absolute error.

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
