# Peer review of "Cross-Validation of Predictive Equation for Cardiorespiratory Fitness by Modified Shuttle Walk Test in Adults with Schizophrenia: A Secondary Analysis of the CORTEX-SP Study"

_ijerph, 2021, doi:10.3390/ijerph182111390_

Round 1

Reviewer 1 Report

Cardiopulmonary exercise testing (CPET), a more realistic option for cardiopulmonary function (CRF) assessment, is time-consuming, requires specialized and expensive laboratory facilities and experts to oversee, and is not widely available in many centers. As a result, patients with schizophrenia (SP) in particular rarely have the opportunity to undergo CRF testing, and as a result CRF is the only major risk factor that is not routinely assessed in clinical practice. In order to overcome the above issues, this study appears to be socially significant in that it creates a new equation for predicting VO2 peak using the Modified Shuttle Walk Test (MSWT), which can be performed at any facility in adult schizophrenic patients.

Some points are noted below for your consideration.

1. As described in the limitation, it is presumed that the prediction equation will change significantly if the patient background is different. Such bias is not likely to be corrected by internal validation (cross-validation). Therefore, if it is claimed that there is better performance than the Singh et al. formula (R2 = 0.57, adjusted R2 = 0.57, P < 0.001), please consider including the patient background used in the Singh et al. study in Table 1, if possible, to prevent misreading by readers.

2. Were there any missing values in creating the prediction equation? If there were, it would be better to specify how to deal with them since they affect the estimated values.

3. In the section on residual analysis of the regression model, P-values are mentioned (outliers, autocorrelation, homoscedasticity), but it would be better to specify which type of test method is used.

4. As stated in the title, one of the key points in this analysis is that 5-fold cross-validation was performed. Therefore, in order to promote the understanding of readers, please consider summarizing the evaluation indices for the five times in a table or figure. In that case, not only R-squared, but also other evaluation indices such as mean square error should be clearly stated so that readers can make a comprehensive evaluation of the prediction equation.

Author Response

Cardiopulmonary exercise testing (CPET), a more realistic option for cardiopulmonary function (CRF) assessment, is time-consuming, requires specialized and expensive laboratory facilities and experts to oversee, and is not widely available in many centers. As a result, patients with schizophrenia (SP) in particular rarely have the opportunity to undergo CRF testing, and as a result, CRF is the only major risk factor that is not routinely assessed in clinical practice. In order to overcome the above issues, this study appears to be socially significant in that it creates a new equation for predicting VO2 peak using the Modified Shuttle Walk Test (MSWT), which can be performed at any facility in adult schizophrenic patients.

Some points are noted below for your consideration.

  1. As described in the limitation, it is presumed that the prediction equation will change significantly if the patient background is different. Such bias is not likely to be corrected by internal validation (cross-validation). Therefore, if it is claimed that there is better performance than the Singh et al. formula (R2 = 0.57, adjusted R2 = 0.57, P < 0.001), please consider including the patient background used in the Singh et al. study in Table 1, if possible, to prevent misreading by readers.

Dear reviewer, after reading your consideration, we have been aware that surely the initial approach to the objectives was not very clear. Thus, the inclusion of Singh et al equation in the results and discussion was not previously well justified in the introduction. Therefore, we have added a sentence in the introduction to justify later the first objective of the study, and the presented results. In this sense, we think that now in both abstract and manuscript the Singh et al study as the original equation is better explained. On the other hand, we have also clarified in the introduction the patient background in the Singh et al study.

  1. Were there any missing values in creating the prediction equation? If there were, it would be better to specify how to deal with them since they affect the estimated values.

Dear reviewer, missing data have not been assumed. Thus, the analysis was carried out by protocol.

  1. In the section on residual analysis of the regression model, P-values are mentioned (outliers, autocorrelation, homoscedasticity), but it would be better to specify which type of test method is used.

We thank the reviewer for this comment. We have now added a paragraph in the statistical analysis section: “In the residual analysis of the regression model, the type of method used for the assessment of outliers was Bonferroni test, for autocorrelation was Durbin-Watson test and for homoscedasticity was Non-constant Variance Score test”.

  1. As stated in the title, one of the key points in this analysis is that 5-fold cross-validation was performed. Therefore, in order to promote the understanding of readers, please consider summarizing the evaluation indices for the five times in a table or figure. In that case, not only R-squared, but also other evaluation indices such as mean square error should be clearly stated so that readers can make a comprehensive evaluation of the prediction equation.

We thank the reviewer for this comment. To perform the cross-validation, the subsamples were randomly selected. For this revision, the cross-validation has been repeated, since the previous version was not saved. Thus, 5 different but similar subsamples were obtained. Consequently, this result would provide greater robustness to the model.

We have now added Table 2 for summarizing the evaluation indices for the five times.

Table 2. The evaluation indices for the 5-fold cross-validation.

Subsets

RMSE

R2

MAE

1

4.617

0.668

3.302

2

4.292

0.681

3.307

3

4.862

0.557

3.757

4

4.196

0.636

3.155

5

4.617

0.586

3.906

1 RMSE, Root mean square error; MAE, mean absolute error.

Reviewer 2 Report

  1. Please change the acronym you use for Schizophrenia to a more suitable form such as SZ or SCZ.
  2. This is a secondary analysis of the original study (Tous-Espelosin et al.. Clinical, Physical, Physiological, and Cardiovascular Risk Patterns of Adults with Schizophrenia: CORTEX-SP Study: Characterization of Adults with Schizophrenia. Psychiatry Res. 2021, 295, 113580). Please clearly indicate this throughout the manuscript including the title, abstract, and method sections.
  3. The authors argue that the study has several strengths, but does not list them. Please explain better.
  4. It would be interesting to include in the final part of the discussion some future prospects and some speculation about the possible clinical fallout of the results presented by the authors.

Author Response

REVIEWER 2

  1. Please change the acronym you use for Schizophrenia to a more suitable form such as SZ or SCZ.

As suggested by the reviewer, we have changed the acronym throughout the manuscript. Schizophrenia (SZ).

2. This is a secondary analysis of the original study (Tous-Espelosin et al.. Clinical, Physical, Physiological, and Cardiovascular Risk Patterns of Adults with Schizophrenia: CORTEX-SP Study: Characterization of Adults with Schizophrenia. Psychiatry Res. 2021, 295, 113580). Please clearly indicate this throughout the manuscript including the title, abstract, and method sections.

Dear reviewer, as you have suggested we have added a clarification in the title, and methods. We have not added any information in the abstract since the number of words is limited to 200.

3. The authors argue that the study has several strengths, but does not list them. Please explain better.

We thank the reviewer for this comment. We have now added strengths. “The current study has several strengths. Considering the difficulties involved in recruiting volunteers with a mental dysfunction, we could argue a relatively large sample (n = 144). Further, the new equation is a very useful and easy tool in the evaluation of this population, and the results obtained with this prediction are better compared to previous studies.

4. It would be interesting to include in the final part of the discussion some future prospects and some speculation about the possible clinical fallout of the results presented by the authors.

Thank you for your suggestion. The following paragraph has been added: “Finally, these results provide evidence for an easy, fast and simple way to evaluate the CRF of people with SZ when objective measurements are not available. This would allow the necessary pre-design assessment of exercise in both clinical and non-clinical settings, and the promotion of exercise programs in this population”.

Round 2

Reviewer 1 Report

The areas I pointed out are getting better.